# Oligo—Not Only for Silencing: Overlooked Potential for Multidirectional Action in Plants

**DOI:** 10.3390/ijms24054466

**Published:** 2023-02-24

**Authors:** Cezary Krasnodębski, Agnieszka Sawuła, Urszula Kaźmierczak, Magdalena Żuk

**Affiliations:** 1Department of Genetic Biochemistry, Faculty of Biotechnology, University of Wroclaw, Przybyszewskiego 63/77, 51-148 Wroclaw, Poland; 2Department of Cellular Molecular Biology, Faculty of Biotechnology, University of Wroclaw, F. Joliot-Curie 14A, 50-383 Wroclaw, Poland

**Keywords:** oligo technology, plant ASO, DNA methylation, gene modulation, epigenetics, gene silencing, gene up/downregulation, non-GMO method, GMO alternative

## Abstract

Oligo technology is a low-cost and easy-to-implement method for direct manipulation of gene activity. The major advantage of this method is that gene expression can be changed without requiring stable transformation. Oligo technology is mainly used for animal cells. However, the use of oligos in plants seems to be even easier. The oligo effect could be similar to that induced by endogenous miRNAs. In general, the action of exogenously introduced nucleic acids (Oligo) can be divided into a direct interaction with nucleic acids (genomic DNA, hnRNA, transcript) and an indirect interaction via the induction of processes regulating gene expression (at the transcriptional and translational levels) involving regulatory proteins using endogenous cellular mechanisms. Presumed mechanisms of oligonucleotides’ action in plant cells (including differences from animal cells) are described in this review. Basic principles of oligo action in plants that allow bidirectional changes in gene activity and even those that lead to heritable epigenetic changes in gene expression are presented. The effect of oligos is related to the target sequence at which they are directed. This paper also compares different delivery methods and provides a quick guide to using IT tools to help design oligonucleotides.

## 1. Introduction

Oligo technology, based on short oligonucleotide sequences introduced into cells to modulate gene expression, is a burgeoning alternative for some aspects of genetic engineering in plants, but it is also a new tool that opens up some unique opportunities for gene control. Oligo technology is the result of exploiting elementary biochemical properties of nucleic acids and naturally occurring gene control mechanisms based on short RNA molecules—RNA oligonucleotides. Genes for such molecules are found in viruses and in cells of both prokaryotic and eukaryotic organisms, where their products exert important regulatory functions.

The mechanisms of action exploited by oligo technology may therefore be similar to those triggered by endogenous sequences. The most important and crucial role of naturally occurring RNA oligonucleotides in cells (especially miRNA) is the tight regulation of gene expression in the context of RNA interference (RNAi). miRNA genes affect the expression of other genes in a manner that depends on the sequence of the mature oligonucleotide they encode. Apart from controlling expression, RNA oligonucleotides are critical to the basic function of DNA replication by serving as primers for DNA polymers.

Genes regulating oligonucleotides are abundant in the vast majority of eukaryotic genomes. They play a key role in homeostasis by establishing the balance between transcripts and proteins in plants and animals. In both kingdoms, miRNA genes are highly conserved and play critical roles in gene regulation [1,2].

The importance of oligonucleotides in cellular functions is underscored by the fact that the last common ancestor of plants and animals already possessed basic components of the miRNA regulatory pathway [3]. The RNA-dependent regulatory machinery of the ancestor of the eukaryotes was composed of proteins from prokaryotes, phages, and archaea. Before these proteins were adapted for the new task of regulating gene expression, their roles appeared to be RNA processing, DNA repair, and protection from foreign or pathogenic genetic material [3,4]. This is also the reason why the RNA-mediated mechanisms of expression control of these two kingdoms show a high degree of similarity. Many researchers believe that the emergence of organisms with highly organized bodies would have been impossible without the sophisticated gene control by miRNA genes, as the explosive evolution of morphological innovations in organisms is highly correlated with the accumulation of new RNA genes that tightly control expression.

However, over millions of years of evolution, plants have developed some additional mechanisms of action (mostly related to DNA methylation) or have begun to use homologous proteins for other purposes. DNA oligonucleotides, which occur naturally in cells, are short-lived byproducts of the digestion of longer DNA molecules belonging to a killed pathogen or to apoptotic or necrotic cells of one’s own body. Underestimated by nature, DNA oligonucleotides have become revolutionary to humans as tools of genetic engineering and biochemistry.

While it is possible to adapt synthesized RNA molecules in oligo technology, such an approach is far-reaching and problematic because RNA is easily degraded (especially in the cell). The very similar structure of DNA oligonucleotides compared to RNA molecules used naturally by cells makes it possible to use DNA oligos as a low-cost substitute. This takes advantage of the fact that they are readily available, cheap to produce, easy to purify, stable, and already used in numerous applications: as primers in PCR reactions, fluorescent and radioactive probes, or drugs.

Nucleic acids (both DNA and RNA) have been used for vector-free modulation of genes (their repair—mutagenesis or modification of their activity) practically since the beginning of the development of molecular biology in both animal and plant organisms. An example of such application is chimeric RNA-DNA oligonucleotides, which are directed to a specific sequence and can cause specific point mutations in this target sequence [5]. Similar techniques have also been used to modify plant genes for example, two teams have independently induced point mutations in the acetolactate synthase gene of tobacco [6,7]. Such technologies for using oligonucleotide sequences 40–200 nt in length to generate small (point) changes/mutations in the sequence of target genes are referred to as oligonucleotide-directed mutagenesis (ODM) and can be successfully used in plants [8,9].

In this work, we focus on a slightly different technology that uses oligonucleotides to induce changes in the activity of plant genes. The main difference between the Oligo technology described here and the ODM techniques is that in the case of the former we do not cause changes in the sequence of the modulated genes, and the expected effect is achieved by inducing epigenetic changes. Moreover, this technology uses much shorter 18–22 nt deoxynucleotides. Exogenous oligonucleotides (20–22 nt), introduced into the cell in the form of single-stranded nucleic acid fragments, are treated by the cellular machinery similarly to those derived from viruses or from the degradation of the transcript by polymerase II. The technology of using oligonucleotide sequences, mostly in antisense orientation (AOS) in mammalian cells, has become the basis of advanced gene therapy techniques for many difficult-to-treat diseases [10]. Its firmly established status has been confirmed by numerous FDA-approved oligo treatments in humans.

Basically, the effect of oligos is not only bidirectional but also affects gene expression at each stage via different types of proteins in the internal regulatory system of the cell, which allows modulation of this process at multiple levels [11,12,13,14,15,16]. In plants, the effects of oligos on the steps of transcription, splicing, translation and DNA methylation are the best documented.

## 2. Oligos Treatment—Alternative Method for Screening of Gene Functions

There is a constant search for new tools for genetic engineering and control of gene expression in organisms, both for basic research and for the development of genetically modified organisms in medicine, agriculture and industry. One of the methods increasingly used for this purpose is oligo treatment. Of particular note is the possibility of DNA methylation modulation by oligos, which leads to heritable changes in gene expression without creating GMO plants [12,17]. Oligo-induced changes are stable and show similar traits to the reference transgenic plants, but without altering the genome sequence [13]. Therefore, oligos provide new tools for plant improvement through noninvasive epigenetic modulation.

Due to the mode of action of oligos, similar to small RNAs, sequence-selective inhibition or enhancement of gene expression enables the elucidation of complex gene expression, especially gene functions and regulatory elements [18,19]. Most importantly, oligos enable the study of vital genes, which is virtually impossible using the classical method of gene knockdown. Gene silencing via RNAi and siRNA is also used for this purpose. Treatment with oligos does not require tedious construction preparation and plant transformation, whereas shRNA and artificial miRNA must be inserted into a plasmid before they can be introduced into the cell, which is difficult and time-consuming [20,21,22]. Furthermore, this means that the degree of inhibition of gene expression depends on the level of expression of the plasmid in the cell. A similar problem occurs when generating mutants with overexpression. Oligos, on the other hand, are dose-dependent and can be used to either increase or decrease expression, depending on the level of expression [23]. At the same time, pleiotropic effects were minimized, which is a common problem when generating mutants by genetic transformation [24,25]. The effects observed after transformation may not only be caused by gene silencing but may also depend on changes in genome sequences and structure caused by the insertion. Unfortunately, this may also lead to altered expression of other genes.

Apart from that, the external addition of oligos enables the study of genes at different stages of plant development and allows experiments to be conducted over time. Therefore, primary and compensatory effects can be distinguished [17]. Regulatory proteins can also be targets of oligos to alter the expression of specific genes or even entire signaling and metabolic pathways. This provides tremendous flexibility in studying gene function and its global impact on hormone balances.

Although small interfering RNAs (siRNAs) can also be delivered directly into the cell, their design and synthesis are more complicated and expensive. Oligos, unlike siRNAs, do not need to be fully complementary to exert an effect [26]. They allow for triggering a change at SNP sites, but because they are inaccurately designed with respect to genome sequences, oligos can lead to expression defects in nontarget genes.

In addition, homologous sequences can be targeted with a single oligo, meaning that a single oligo can inhibit more than one gene from the same gene family [27]. Important in the context of the applicability of oligo technology is the ability to target regulatory proteins, which allows the study and modulation of entire signaling pathways and the study of the influence of individual factors on cell function.

## 3. Mechanisms of Action

In general, the action of exogenously introduced nucleic acid sequences (oligo) can be divided into direct interaction with endogenous nucleic acids (genomic DNA, hnRNA or transcript) and indirect interaction via induction of processes regulating gene expression (at transcriptional and translational levels) involving regulatory proteins using endogenous cellular mechanisms [28]. A diagram showing the putative mechanisms of action of oligonucleotides on gene activity can be found in Figure 1.

It is usually assumed (mainly on the basis of tests on animal cells) that the introduction of short oligodeoxynucleotides into the cell leads to hybridization of oligos with the homologous region in the transcript sequence (direct interaction). This event activates RNA-dependent RNA polymerase (RdRP) and leads to the formation of double-stranded RNA (dsRNA) that may be cleaved by DICER-like proteins (DCL) and can be integrated into AGO4 or AGO6 proteins, which might be responsible for gene repression as an effect of RNA interference (RNAi) or can also activate particular genes in the process called RNA activation (RNAa).

The probable mechanism whereby this occurs is via RNase H1 or RISC pathways that reduce gene expression (this process, mainly using animal cells as an example, was excellently described in [29]), but can also increase gene expression through modulation of splicing or translation or stability by protecting AU-rich element (ARE) of mRNA [30]. The RNase H pathway is unique to DNA oligo as opposed to the natural mechanisms induced by gene-regulating RNA molecules. It was also demonstrated that antisense oligonucleotides can reduce mRNA levels by acting through the no-go decay pathway. It is also possible that the degradation of polyA is induced by the presence of the complementary oligo. This would lead to rapid digestion of an unstable (due to lack of polyA) transcript.

Splicing of pre-mRNA, a dynamic process in which introns are removed and exons are joined, is governed by a combinatorial system exerted by overlapping cis-elements unique to each exon and its flanking intronic sequences. Oligonucleotides (mainly antisense oligos (ASOs)) can block splicing cis-elements of splicing and/or affect RNA structure and modulate splicing in vivo [31].

The next step in the expression of the gene is the step of translation. It has been shown that after base pairing with the target RNAs, oligonucleotides can recruit RNase H1 to cleave the RNA substrate within the region complementary to the oligo. RNase H1, which is expressed at low levels in all cells and localized in both the nucleus and cytoplasm, seems to be a limiting factor with respect to ASO-mediated antisense activity [15,32]. Oligonucleotides can efficiently reduce the levels of both nuclear and cytoplasmic RNAs. Results achieved on animal cells show that many oligos can rapidly reduce levels of cytoplasmic mature mRNAs without affecting the levels of nuclear pre-mRNAs [15]. There is no confirmation of this phenomenon occurring in plants so far.

RNase H1-dependent oligonucleotides can trigger rapid degradation of mRNAs in the cytoplasm, where most mRNAs are translated under normal conditions. It is therefore possible that oligos can act on translating mRNAs. In such cases, the activity of oligos may be affected by the translating ribosomes. Translating mRNAs are being rapidly scanned by one or more ribosomes per mRNA. Scanning ribosomes can actually remove ASOs from the mRNA before RNase H1 is recruited, resulting in altered ASO activity [14]. The activity of oligonucleotides on mRNAs should depend on several rates: the on/off rate of oligo binding, the rate of RNase H1 recruitment, the rate of RNase H1 cleavage, and the rate of translation. Additionally, formation of oligonucleotide-target hairpin complex involves some type of triple-stranded structure with noncanonical interaction (Hoogsteen hydrogen bonds), therefore leading to more specific recognition and a higher affinity of the bond.

The above mechanisms are described for both animal and plant cells [33,34]. Other mechanisms could be responsible for oligo activity, including direct binding to genomic double-stranded DNA and triplex DNA generation. This conjecture is based on our original finding that oligo affects genes and the methylation status of the genome [12,13,17]. Triple helix-forming oligonucleotides can compete with the binding of transcription factors and affect transcription initiation or elongation [35]. Recently, selected CCGG motif has been shown to be differentially methylated in response to plant treatment with oligo [12]. For the time being, there is no clear explanation for this. One hypothesis states that there is a distinction between complexes containing antisense or sense-oriented oligos, with the biggest difference between molecules of both types being the strength and stability of the effect on gene expression.

Further, it has been found that there is a difference between complexes containing antisense or sense-oriented oligo. Not only may sense or antisense oligo induce opposite effects on gene expression (up- or downregulation), but can also lead to different strengths of expression changes.

As for mechanisms operating on the genome level (epigenetic modulation) in plants, it seems that the mechanism tends to be based on DNA methylation, whereas in mammalian cells it is associated with histone modification [36]. DNA methylation is considered to be one of the most important epigenetic marks in plants [36]. Attachment of a methyl group in the 5‘ position of the DNA cytosine (5-mC) and, as was recently indicated, 6’ position of the DNA adenosine (6mA) are important epigenetic modifications in plants [37]. The process of addition or deprivation of a methyl residue is based on the recruitment of the methyltransferases—in plants by RdRP or polymerase IV and polymerase V complex [38]. Changes in nucleic acid methylation state have been reported to have a robust impact on plant phenotype. It has been shown that transgenic rice (*Oryza sativa*) and potato (*Solanum tuberosum*) with altered methylation gave 50% higher yield, biomass and increased resistance to drought stress [39].

Research on flax plants shows that it is possible to obtain both repression and overexpression of the target gene using oligo technology. The mechanism behind overexpression that has been proven is the induction of changes in genomic DNA methylation (most frequently CG sites but also CHG, CHH) leading to the reorganization of the chromatin structure via shift of the nucleosome position [12].

Recently, the possibility of inheriting oligo-induced epigenetic changes and thus changes in gene activity for at least 3 generations has been demonstrated [13,17]. By inducing epigenetic changes (mainly changes in genomic DNA methylation), it was possible to produce a new variety of linseed—Silesia [40]. Tests for distinctness, uniformity and stability (DUS) and value for cultivation and use (VCU) were carried out in the experimental stations of the Research Center for Cultivar Testing (Polish name COBORU). The result was the registration of the first epigenetically modulated variety in the National List and the granting of the breeder’s right for this variety (the status was obtained in March 2020) and would subsequently lead to its registration in the Common Catalog EC. The described experiments were carried out on flax, and it seems very likely that obtaining similar effects for other plant species is possible. The strongest effect of stabilization of changes was obtained via oligos that caused changes in methylation of certain regions of the genome. Thus, it appears that inducing changes in methylation, a key epigenetic mechanism for plants, may be a guarantee of permanent, inherited changes.

## 4. Oligo Design Based on the Relationship between Sequence and Effect on Gene Expression

### 4.1. Basic Rules of Design

As mentioned above, the use of oligonucleotides in plant research can be considered to be at an early stage. There is much more knowledge and experience in the field of using oligonucleotides in mammalian research or therapy. This is also true for a technical topic like oligonucleotide design.

Homologues of plant epigenetic factors can be identified in animal organisms. In eukaryotes, high concordance of epigenetic mechanisms and their evolutionary conservation can be observed [41]. This implies the possibility of adapting the rules for the evolution of oligonucleotides acting on mammals for the evolution of oligonucleotides for plants.

A 15-nucleotide antisense fragment is long enough to make a specific association with the target mRNA sequence. The optimal length of an oligonucleotide is 15–20 [23] or 15–30 [42] nucleotides.

A critical feature defining the performance of the oligo is its complementarity with the target fragment. There are conflicting reports on the specificity of the oligo (with respect to the target sequence). Xie et al. [43] state that even two-point mutations can limit the effect of the oligo. This possibility is also confirmed by the results obtained with the gene CAB [11]. However, other results concerning modification of two very homologous isoforms of the CHS protein in flax [12,13] do not confirm this.

One of the most important conditions for the efficiency of an oligodeoxynucleotide is that it can hybridize with the target fragment of the nucleic acid [23]. Therefore, the most important task in oligo design is to define the secondary structure of the target nucleic acid as precisely as possible.

The appropriate fragment is free of structures that prevent steric hybridization [23,44]. Currently, a major facilitator in this design aspect is the increasing availability of fully sequenced genomes and the ability to use them as input data in IT tools [27].

Certain secondary structures may favor hybridization of oligonucleotides. End-strand fragments, single-stranded loop fragments of harpins, and common sequences longer than 10 nucleotide beads can be good targets [44,45,46]. Base composition should also be considered when designing antisense oligonucleotides. CCAC, TCCC, ACTC, GCCA, and CTCT are motifs whose presence correlates positively with oligonucleotide efficacy. In turn, the motifs GGGG, ACTG, AAA, and TAA are characterized by a negative correlation [47]. Higher efficiency is obtained with ASOs containing at least 11 G or C per 20 nucleotides [48].

### 4.2. IT Tools

IT tools come in handy when it comes to designing antisense oligonucleotides. Mfold, currently accessible from the UNAFold (http://www.unafold.org, accessed on 5 December 2022) domain, developed in 2003 [49], enabled prediction of all optimal or close to optimal structures of a given mRNA using minimum free energy (MFE) prediction algorithm. Mfold’s input is a clear mRNA sequence or in FASTA format as well as parameters such as folding temperature, ionic conditions and constraints of pairing particular bases. It is also possible to determine graphic and file format properties. During query creation, access to documentation is available. The obtained output includes a folding energy dot plot, several predictions of structures, their graphical representation and thermodynamic details.

Another program available on the web is Sfold (https://sfold.wadsworth.org, accessed on 1 December 2022), which uses other mathematical algorithms based on stochastic sampling in Boltzmann ensemble and clustering to obtain the best predicted secondary structure of mRNA [50]. Furthermore, hybridization probability of antisense oligonucleotide in specific sites of predicted structure is calculated. Sfold facilitates selecting optimal oligo considering empirical rules like GC content and avoidance of GGGG motifs. Outputs can be displayed in graphical or text format and contain information about predicted binding energy. Therefore, the use of both programs is critical for the optimal design of oligos, giving the chance to achieve the best effect [44].

Let flax (*Linum usitatissimum* L.) chalcone synthase (Phytozome database, CHS1—Lus10031622, 1194 bp) transcript analyzed by mFold and sFold be an example output for these IT tools. Appendix A shows respectively, energy dot plot and one of proposed folding (deltaG = −371 kcal/mol) of CHS1 mRNA generated using mFold 2.3 with default parameters. SFold output (also default parameters) Appendix A provides a probability profile of CHS1 displaying predicted accessible sites on the target RNA. It is also possible to receive a probability profile of a specific target fragment. Accessible sites can be targeted by several oligos. It is important to select optimal ones. Text output for antisense oligos is a representation of proposed oligos covering the target sequence iterated by one base and their determined parameters. Figure 2 presents a fragment of filtered output of antisense oligos for CHS1. Selected oligos meet the conditions of binding energy (oligo with a lover binding energy is preferable) and rules like GC content (40–60%) or GGGG motif absence.

Cellular factors involved in determining mRNA accessibility are still not well understood. Despite continuous improvement and the possibility of applying comparative methods using several available algorithms, precise prediction of secondary structures remains troubling [45,50]. Therefore, experimental validation of the effectiveness of the designed oligo is essential [12]. In order to avoid aspecific interactions, it is recommended to perform homology analysis on a target organism’s mRNAs using available databases. Great attention should be paid to the use of appropriate controls to prove that the effects (changes in expression) result from the impact of a specific oligo on the target [23].

### 4.3. Oligo Action Based on Target Sequence

Effect of the oligo—overexpression or suppression—depends strongly on the target sequence to which it is directed. It is possible to distinguish specific target sites for antisense oligonucleotide activity such as exons, 3′ and 5′ UTR regions, and promoter regions: 100% of oligos targeting the “core promoter,” a region located ∼40 bp upstream of the transcription initiation site, result in upregulation of the gene [12]. Upstream of the core promoter region are the proximal and distal regions of the promoters (up to 1000–1500 bp upstream of the transcription initiation site). Proximal and distal regions of the promoter contain various regulatory sequences such as enhancers, silencers, insulators, and cis-elements that contribute to the fine regulation of gene expression at the transcriptional level. Alignment of the oligonucleotide to the 3′ end can lead to cleavage of the poly-A tail and consequent degradation of the mRNA. Negative regulatory elements may be present in the 5′UTRs of some mRNAs. Work at the 5’ end or at the translation initiation site can result in hindering the assembly of the translation system [46]. It has also been shown (in mammalian cell cultures) that translation can be enhanced by antisense oligonucleotides (ASOs) targeting upstream open reading frames. CHS gene study shows that oligonucleotides targeting the 5’UTR (very close to the start of translation) cause overexpression [12].

Immature mRNA can serve as a target for antisense oligonucleotides. Targeting the splicing site may prevent the correct assembly of the mRNA [51]. An oligo targeting exons can have a differential effect, mostly repression (67% of OLIGO analyzed), but also slight (less than in the case of targeting promoter sequences) gene activation or no effect, as oligo targeting intron sequences repression [12,13].

One of the major challenges associated with the use of antisense oligo is its stabilization, as the natural molecule of oligo–deoxyribonucleic acid is subject to rapid endonucleolytic and exonucleolytic degradation once introduced into the cell. Oligo modifications are much more popular in experiments with animal and mammalian cells, but especially in human therapy. To prolong the operation and availability of oligos, chemical modifications are made to the molecules. Modified oligos are expected to be eliminated from tissues more slowly, to increase the percentage of drug in tissues, and to be lower in dose and/or administered less frequently. The stability of ODNs can be increased by thiophosphorylation (PTO), in which the nonbridging oxygen atom in the phosphorothioate backbone of oligo is replaced with sulfur. The PTO modification is particularly popular because it provides sufficient stabilization against nucleolytic degradation while allowing recognition by RNase H [23]. In the second common modification—MOE—a change occurs at the 2’-position of the ribose units (2’-methoxyethoxy), resulting in greater nuclease resistance and higher binding affinity. Unlike PTO, MOE oligonucleotides are not subject to RNase H action. Therefore, they act by blocking the translational machinery [23,52].

In experiments with plants treated with modified oligo (PTO or methylation of all cytosines in sequence), significantly higher stability of epigenetic changes was observed in successive generations of generative plants. However, a pronounced induction of genes involved in phenylpropanoid metabolism (which was not targeted by the oligo) was observed in plants treated with oligo PTO. This could indicate a defensive response of the organism due to the potential toxicity of the exogenous factors used. No effect was observed with methylated oligo (less stable than PTO) [13]. Moreover, progeny obtained from seeds of plants treated with oligo PTO were significantly less vigorous and produced very few seeds of their own.

## 5. Methods of Introducing OLIGOs into the Plant Cell

Oligo technology is less commonly applied to plant cells than to animal cells, but it has already been shown to work in several species: flax, barley, tobacco, tea and recently cucumber and potato [10,11,12,14,15,17,34,53]. This technique was first successfully applied in plant cells to alter the expression of the gene encoding the transcription factor SUSIBA2 [24]. The researchers’ results showed that antisense oligonucleotides were efficiently transported within the leaf and reached the nucleus and chloroplasts [11,24]. In general, plant cells are more susceptible to oligo treatment than animal cells. This is mainly due to the positively charged cell wall, which is not a barrier for the negatively charged oligonucleotide molecules. Moreover, oligonucleotides can enter plant cells through channels specific to sugar molecules [24]. The exact mechanism of import into a plant cell has not been fully described.

Oligonucleotides can be introduced into plant cells in several ways: infiltration under reduced pressure, infiltration through the stomata, spraying of cells, forced osmosis, and biolistic particle delivery system (gene gun) [54]. Depending on the plant species, tissue type and age (developmental stage), different delivery methods are most suitable. For details, please see Table 1.

## 6. Analysis Strategy for Oligo Actions

Oligos are usually designed for a specific target. Therefore, appropriate validation allows determining which one works “best,” according to our expectations. Based on proper design of oligos, we can expect that they will act in a certain way on the target gene or sequences [12,16]. Nevertheless, not all aspects of oligos have been fully examined or explained yet. Therefore, it is necessary to determine at which stage of the gene expression oligo acts (e.g., transcription or translation) and the direction of those changes (down or up). For this purpose, it is recommended to check both the target transcript (e.g., by qPCR), and if it results in a protein product, also their value (e.g., by Western blot). In some cases, the enzymatic activity or the level of final metabolites are also determined [11,17,55]. Moreover, fluorescence labeling of oligos enables the determination of their subcellular localization. It is especially important when the target sequence is located outside the nucleus, e.g., in chloroplasts [11], which allows determining whether the oligos have reached their destination.

In plants, oligos have been shown affect DNA methylation, which can modulate the expression of the target gene. However, oligo usage may also lead to methylation modulation throughout the genome. The resulting epigenetic changes can be stable for up to 3 generations. Therefore, the total degree and profile of methylation would be worth checking with particular regard to the target gene [13].

## 7. Summary

The advent and development of methods in genetic modification of organisms have revolutionized science, medicine, pharmaceuticals, agriculture, and the chemical industry. To date, seven antisense therapeutics have received regulatory approval and dozens are in clinical development [18,56,57,58,59]. One of the FDA drugs—milases—is a patient-tailored antisense oligonucleotide drug for Batten disease. A successful example of this therapy may indicate that antisense oligonucleotides are useful in developing individualized treatment [60]. The discovery of the principles of epigenetics and the technologies resulting from new knowledge in this field are opening up innovative ways to better adapt living organisms to human needs. GMO technology, which is heavily used in plant biotechnology, has difficulty commercializing GM plants and products due to community disapproval and regulatory issues. Oligo technology faces the challenge of being a GM alternative by modulating the genome rather than altering it. The oligo method undoubtedly has great potential benefits and is also attractive because of its low cost, high efficiency [22], and the flexibility associated with the possibility of adapting the application method/system/mode to the characteristics of the target plant. A great advantage of oligo technology is also the possibility of using it for both dicotyledonous and monocotyledonous plants, which cause difficulties with the classic transformation with *Agrobacterium*.

Most studies using oligonucleotides (mainly antisense) are performed to obtain transient transformants and to study gene function. To date, there has been little attempt to stabilize the changes induced by oligonucleotides. As far as we know, there is only one case in which a crop variety was generated using oligo technology [13,40]. However, there is a chance that by developing the knowledge of stabilizing epigenetic changes in plants, the changes in gene activity induced in this way will be inherited. This will make it possible to use this method for crop improvement as an alternative to genetic modification.

The use of this method is further facilitated by the development of accompanying fields such as genome sequencing and in silico techniques (IT tools). The diversity and multistep nature of the mechanisms of action of oligonucleotides and the differentiation of effects in direction and strength offer a wide range of potential applications. Reports of inheritance of epigenetic changes induced by the action of oligos promise efficient practical application of this method in areas such as agriculture, protection of crops from infections, and improvement of plant adaptability to stress and climate change [34,53]. The use of oligonucleotides in research should not be underestimated. Their application in plant tissues can open the way for high-throughput screening for gene function [61]. There is still much to be explored in the field of oligonucleotide use, especially with regard to application in plants. The key challenge appears to be minimizing the occurrence of unanticipated effects and more accurately predicting the changes caused by the oligo used. Oligo technology offers many advantages while not requiring a large investment of time and money.

## Figures and Tables

**Figure 1 ijms-24-04466-f001:**
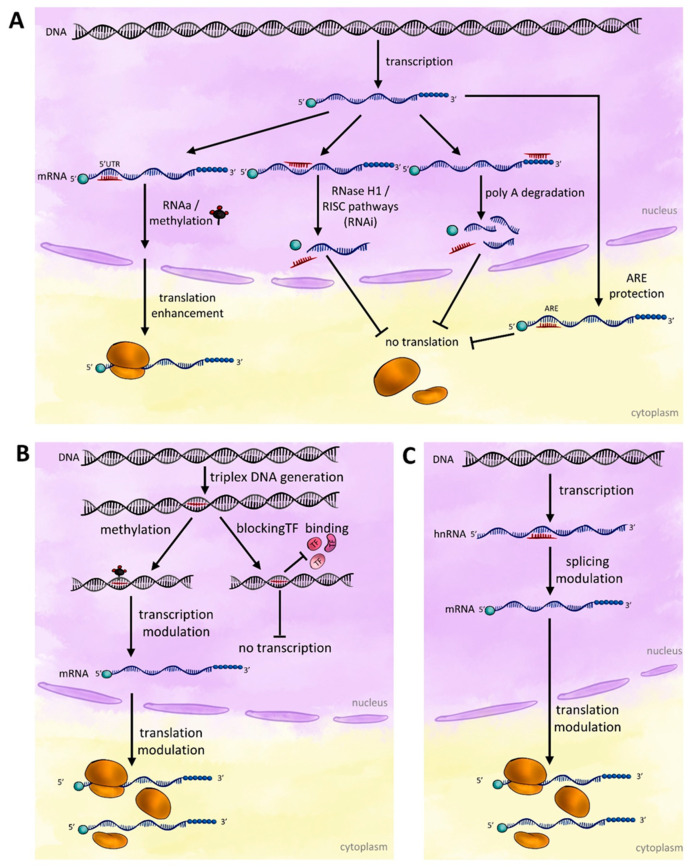
Possible mechanisms of action of oligos in the plant cell depending on the site of binding: (**A**) interaction at the level of genomic DNA, (**B**) at the level of transcript maturation (hnRNA), (**C**) interaction with the transcript (mRNA). Please refer to the main text for further details.

**Figure 2 ijms-24-04466-f002:**
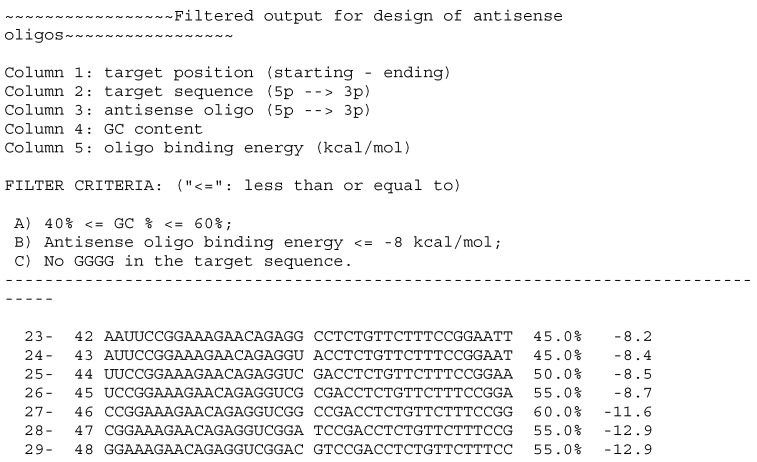
Fragment of example text Sfold output for CHS. Filtered oligos that meet designated criteria showed with their start—ending target position, energy and GC content parameters.

**Table 1 ijms-24-04466-t001:** Method for delivering an oligonucleotide to a plant cell.

Method	Tissue Type/Developmental Stage	Advantages	Disadvantages
infiltration under reduced pressure [11,12]	almost any type of tissue and any stage of development: whole plant, leaves, roots	quick and easy to perform method, many plants can be infiltrated at the same time, spiking is possible to a certain extent (easy to seedlings)	considerable stress, a large volume of oligonucleotide solution is needed at once and tissue residues contaminate the tissue, it is difficult to transfer the method to adult plants
infiltration through the stomata [53]	leaves only (the larger the better), best results with mature leaves, possible study in vivo	small volumes of the oligo solution required at once, no problem with impurities, no equipment required	more variable results, possibility of tissue damage, more difficult to do manually, especially troublesome with some small-leaved species such as flax
spraying the cells/tissue [43]	different developmental stages and organs, possible life examination for above-ground part of the plant only	simple scale-up, hardly any equipment required	more variable results, more difficult to control quantitative application of oligo
uptake in a sugar solution [17,24,25,43]	whole plant, leaves, roots, best results with seedlings, possible study in vivo	quick and easy to perform method, hardly any equipment re-quired	large volume of oligonucleotide solution is needed at once, it is difficult to transfer the method to adult plants

## Data Availability

Not applicable.

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
