# Peer review of "Oligo—Not Only for Silencing: Overlooked Potential for Multidirectional Action in Plants"

_ijms, 2023, doi:10.3390/ijms24054466_

Round 1
Reviewer 1 Report (Previous Reviewer 3)
Author improved my comments and now paper can be accepted.
Author Response
Thank you for your time and valuable comments that helped us improve our manuscript.
Reviewer 2 Report (Previous Reviewer 2)
Dear authors,
In my second review of your paper, I saw that you took most of my comments and suggestions into account. However, some aspects still seem to have been overlooked. I have pointed out them as follows;
In L. 226,227-303 (…that transgenic rice (Oryza sativa) and potato (Solanum tuberosum) with altered …) and (Let flax (Linum usitatissimum L.) chalcone ..): Latin names should be written in italic form.
In L.386 (…cific for to sugar molecules …..): "for" should be replaced with "to".
In L. 406 (…which allows to determineing….): It had better use the “ing” form of "verb" after "allow".
In L. 407 (..whether the OLIGOs had have reached its their destination …): In the phrase, changes should be performed as shown.
In L 414-415, (The advent and development of ……….. industry.): If the first sentence of Summary is changed to “The advent and development of methods in genetic modification of organisms have revolutionized science, medicine, pharmaceuticals, agriculture, and the chemical industry.”, like it will get better.
Author Response
Thank you for your time and valuable comments that helped us improve our manuscript. We have corrected the article according to your suggestions. Once again, thank you very much.
This manuscript is a resubmission of an earlier submission. The following is a list of the peer review reports and author responses from that submission.
Round 1
Reviewer 1 Report
Cezary Krasnodębski et al report the "Oligo-not only for silencing. Overlooked potential for multi-directional action in plants". The subject of the review is interesting, and the manuscript is well structured. The analysis logic is appropriate, and enough detailed references are provided to support their review. This overview enables us to better understand the basic principles of oligo action in plants. I suggest that this review can be received directly.
Reviewer 2 Report
Dear authors,
The paper felt like it was handled too quickly. Many typos and spelling errors have been identified. These are marked on your pdf. Please find it in an attached file. When you take a look at them, you will notice.
More importantly, you used first-person plural expressions as if you were writing a research paper and discussing your findings. In addition, there is an author who prepared the review in the study you call our own research on flax plants. It can turn into a conflict of interest. I would like to point out that such expressions are not appropriate for a paper.
Although there are some reviews (e.g. Beetham, et al., 1999; Hammond et al., 2005; Gocalet al.,2015; Sauer et al., 2016; Songstad et.,2017; Wdowikowska et al.,2021 so on…) on this subject, it has not been adequately explained why such a review is needed. Furthermore, these studies were not cited as references.
So, thought that its originality and fiction are not sufficient for international journals. As such, it addresses national journals and scientific magazines rather than international journals.

Reviewer 3 Report
The MS is overall excellent and interesting. But the major drawback is that the authors provide significantly less information about the topic. I suggest authors add more details on the application of oligo in plant improvement. Add discussion about recent work that has been done on this topic.